# Effects of Ambient Temperature and Humidity on Natural Deposition Characteristics of Airborne Biomass Particles

**DOI:** 10.3390/ijerph20031890

**Published:** 2023-01-19

**Authors:** Ye Yuan, Shuo Li, Tiancong Chen, Jianlin Ren

**Affiliations:** School of Energy and Environmental Engineering, Hebei University of Technology, Tianjin 300401, China

**Keywords:** biomass particle, deposition, temperature, relative humidity

## Abstract

In the production process of biomass energy with crop straw as the raw material, the indoor dust environment created by smashed plant fiber can affect the health of workers and lead to the risk of fire and explosions. The physical properties of biomass vary with the ambient air conditions, resulting in different deposition processes for airborne biomass particles. In this study, the deposition of biomass particles in different environments in an experimental chamber was examined by independently controlling the internal temperature and relative humidity. The results show that in the ambient temperature range of 20~40 °C and at a relative humidity of 25~65%, the water absorption rates of the biomass particles were 15.4~24.7%. The deposition rates of the airborne biomass particles with different sizes were 0.9~2.9 h^−1^, which positively correlated with the particle sizes in the same ambient conditions. The increase in ambient temperature and relative humidity promoted the deposition of biomass particles with diameters over 0.5 μm. For the particles with diameters below 0.5 μm, the deposition rates were nonlinearly related to the ambient temperature and relative humidity and were greater at lower temperatures. The significance levels of the factors influencing the particle deposition were particle size > ambient temperature > ambient relative humidity. For the biomass particles below 0.5 μm, the influence of the relative humidity on the deposition was much weaker than that of the temperature.

## 1. Introduction

Recently, with the advocacy of and support for renewable energy utilization, biomass energy made from crop straws is undergoing wide development and application in China [1,2]. Due to their various shapes, crop straws often need to be crushed in the energy production process [3]. These crushed straws easily form dust in their production process, which includes crushing, transportation, loading, unloading, etc. The dust formed by plant fiber particles not only affects the health of operators [4] but also increases the risk of fire and explosions [5,6]. In addition, the particles deposited in the internal gaps of the equipment may result in mechanical failures [7]. Therefore, the production of biomass energy must include a focus on the control of indoor airborne particles.

Biomass particles are suspended in the indoor factory space, forming a gas–solid two-phase field. Deposition refers to particles settling on the surfaces of objects and space due to gravity, diffusion, and other effects; it is affected by many factors, such as the properties of the particles and the characteristics of the ambient environment. The enhancement of particle deposition is a common method to reduce aerosol concentration. The deposition of indoor particles under different conditions has been widely researched.

Zhao et al. [8] investigated indoor particle deposition in four commonly used ventilation modes by using simulations and experimental tests. The results showed that ventilation plays an important role in determining particle deposition due to the obvious differences in particles’ spatial distribution. Lee et al. [9] estimated the particle deposition rate under different air exchange rates, and they found that increasing the air exchange rate was beneficial for indoor particle deposition. Xiao et al. [10] tested the particle deposition in an enclosed chamber, and the results showed that the deposition rate increased with the air temperature. Ren et al. [11] studied the deposition of airborne particles on the surfaces of clothing and found that ambient particle concentrations and spatial surface characteristics had significant effects on the deposition of the particles onto the surfaces of clothing. Kim et al. [12] investigated the effects of flow and humidity on particle deposition in an enclosed chamber. The results implied that an increase in ambient relative humidity can significantly increase particle deposition.

From the existing research results, a theoretical basis and empirical conclusion have been provided for studying the motion characteristics of airborne biomass particles in indoor space. Due to their special physical properties, such as water absorption and morphological variability, the behavior of suspended biomass particles is easily affected by the ambient temperature and humidity [13,14]. In fact, the indoor environments of biomass energy plants vary with different production processes [15]. Taking the biomass briquet plant as an example, some of the moisture contained in the biomass materials is evaporated by the friction heat in the briquetting process, forcing the local environment near the briquetting machine into high heat and high humidity. The non-homogeneity of indoor environments can affect the deposition of biomass particles, leading to different particle concentrations in different indoor spaces.

In this paper, an experimental chamber with independent temperatures and relative humidity control is built. Based on the physical properties of biomass particles and the indoor environmental characteristics of biomass energy plants, the natural deposition characteristics of biomass particles at different ambient temperatures and relative humidity levels are examined.

## 2. Materials and Methods

The physical properties of biomass particles are affected by raw materials, crushing mechanisms, and the environment. In this paper, a biomass briquette production plant in northern Hebei Province, China, is taken as an example. The raw material for briquetting in this plant is the straw of naked oat, a common crop in the local region. The indoor airborne particles are emitted mainly in the briquetting process, with hot vapor as the driving force. The particles of naked oat straw were used as the materials for experimental research.

### 2.1. Materials and Experimental Conditions

In order to simulate the airborne particles in the briquetting plant, the raw straws of naked oat were collected in the plant and ground by the CJM–SY–B multi-dimensional vibrating ball mill pulverizer (Qinhuangdao Taijihuan Nano Products Co., Ltd., Qinhuangdao, China). The size distribution of the ground particles was analyzed by the Malvern MasterSizer2000 laser particle size analyzer (Malvern PANalytical, Malvern, UK), as shown in Figure 1. It can be seen that the volume fractions of the biomass particles with different diameters showed a normal distribution. The median and maximum particle diameters were 9.6 μm and 247.2 μm, respectively. The average diameter was 0.65 μm.

The indoor environment measurement of the briquetting plant’s production status shows that under different production conditions, the indoor temperature varies between 14.6 °C and 28.2 °C, and the indoor relative humidity varies between 21.4% and 61.3%; around the briquetting machine, the local air temperature varies between 18.4 °C and 33.5 °C, and the local relative humidity varies between 50.4% and 69.4%. According to these measurement results and considering the feasibility of the environmental control of the experimental chamber, the experimental temperatures and relative humidity were set at 20~40 °C and 25~65%, respectively. Taking 5 °C and 20% as the control difference, the overall experimental conditions were divided into 15 groups.

### 2.2. Building and Setting of the Experimental Chamber

Based on the purpose of this research and referring to the experiences reported in related experimental research [16], an experimental chamber was built, as shown in Figure 2. The volume of this chamber was 1 m × 1 m × 1 m. To prevent electrostatic influence on particle movement, the inner walls were built using antistatic polycarbonate plates, which were wrapped in polystyrene extrusion plates for heat insulation. To regulate the temperatures of the wall and internal air, graphene heating films (thermal power density of 400 W/m^2^) were embedded between the two layers, and two air heaters (each with a power of 1000 W) were installed on the chamber ceiling. An ultrasonic humidifier (adjustable humidification capacity of 1500 mL/h) was connected through the back wall of the chamber to regulate the internal relative humidity. Two axial fans (power 35 W, speed 50 rpm/s) were installed on the chamber ceiling to quickly make the internal particles and ambient parameters even. The front wall of the chamber could be opened with a particle injection port at the center. All the joints, valves, and openings of the chamber were sealed for airtightness during the experiment. After commissioning, the temperature and humidity in the experimental chamber could be uniformly controlled in the range of 20~70 °C (≤±1 °C), RH = 20~70% (≤±3%).

To monitor the internal particle concentration, a Colnot GT–1000 laser particle size analyzer (as shown in Figure 2c) was fixed in the center of the chamber floor. Near the spatial center of the chamber, two air temperature and humidity measuring points were arranged by applying a DWL–20E temperature and humidity recorder (Yowexa, Shenzhen, China). Four T-type thermocouples were uniformly distributed on each side of the internal walls to monitor the wall temperature. The key parameters of the measuring devices are shown in Table 1. During the experiment, the temperatures of the chamber walls and the internal air were kept constant and the same. Therefore, after the internal temperature and humidity were set, the internal moisture would not condense into droplets, which would not affect the accuracy of the laser particle size analyzer.

### 2.3. Parameter Setting and Process Design

In addition to deposition, coagulation is another key process affecting indoor particle concentration. Results from relevant research show that coagulation between particles is negligible when the particle concentration is less than 3 × 10^3^ p/cm^3^ [17]. The measurement in the briquetting plant shows that the average mass concentrations of indoor PM_2.5_ and PM_10_ are 266 μg/m^3^ and 310 μg/m^3^, respectively. Considering the size distribution of the biomass particles, as shown in Figure 1, in each group of experiments, the mass of the particles injected into the chamber was controlled at about 0.01 g.

The experiment featured 15 working conditions, each of which was repeated three times. For the briquetting machine in the example plant runs, which lasted about 20~30 min each time, the deposition time of the biomass particles in the chamber was set to 30 min. With the temperature gradient near the wall, thermophoresis can influence particle deposition [18]. To analyze the particle deposition in large indoor spaces under different environmental conditions, thermophoresis in the experiment chamber should be eliminated. Therefore, the temperatures of the inner walls and internal air are kept the same during the experiment. The detailed process of the deposition experiment is as follows:(1)Turn on the laser particle size analyzer and the temperature and humidity recorder, then close the chamber. Heat the chamber walls and the internal air with the graphene heating films and air heaters, while adjusting the internal relative humidity with the humidifier. Turn on the axial fans to accelerate the uniform distribution of temperature and humidity in the chamber.(2)After the internal air temperature and relative humidity are stabilized at the set conditions, turn off the air heaters and humidifier. Reduce the power of the graphene heating films to keep the wall temperature consistent with the internal air temperature. Inject 0.01 g of biomass particles into the chamber through the injection port, with the axial fans kept running for 2 min. Next, turn off the axial fans to let the particles deposit naturally in the chamber. Set the moment when the fan is turned off as *t* = 0.(3)After the deposition has lasted for 30 min, export the data of each instrument and clean the chamber.(4)Switch the experimental conditions and repeat the above steps.

The hygroscopicity of biomass materials can affect their physical properties, such as density [19], and the shapes and sizes of biomass particles may vary in different ambient environments, resulting in different deposition characteristics [20]. To further analyze the impact of humidity on the deposition process of biomass particles, the physical properties of the biomass particles in different experimental conditions were examined in addition to the deposition experiment. The conditions of low-temperature–low-humidity (20 °C, RH = 25%) and high-temperature–high-humidity (40 °C, RH = 65%) were taken as representatives. The morphology and water absorption of biomass particles in these representative conditions were examined by the following steps:(1)Place the biomass particle samples in a thermostatic dryer (DH–101–1BS, Tianjin Zhonghuan Experimental Electric Furnace Co., Ltd., Tianjin, China) for 4 h, then cool the samples to ambient temperature in a dryer. Meanwhile, regulate the internal temperature and relative humidity of the experimental chamber to the representative condition.(2)Divide the dried particles into two parts. Observe one part with a 7610F scanning electron microscope and weigh the other part with an electronic balance (PRACTUM224–1CN, Dolis Scientific Instrument (Beijing) Co., Ltd., Beijing, China).(3)Spread the weighed particles evenly on a 20 cm × 20 cm glass plate. Place the glass plate (with particles) into the commissioned experimental chamber and seal the chamber.(4)After 30 min, remove the glass plate and collect the particles. Weigh the collected particles with the electronic balance. Calculate the water absorption rate by using the following equation:(1)$$X=\frac{m_{1}-m_{2}}{m_{2}}$$
where *X* is the water absorption rate of the environment-treated biomass particles; *m*_1_ is the original quality of the dry particles, g; *m*_2_ is the particle quality after treatment in the chamber.(5)Observe the morphology of the treated particles by the scanning electron microscope (SEM).

### 2.4. Data Processing Method

To reflect the deposition intensity at different concentrations, previous research on particle deposition mostly focused on the deposition rate [21,22,23]. In a confined space in which the particles are uniformly mixed and the deposition process is the only influence on the particle concentration, the variation of the particle concentration can be described as follows:(2)$$\frac{dC}{dt}=-\beta C$$
where *C* is the particle concentration, p/m^3^; *t* is the time moment, h; and *β* is the deposition rate, h^−1^.

Through the integral form of the above equation, the deposition rate *β* can be calculated with the particle concentration at any time:(3)$$\beta =-\frac{1}{t}\ln\frac{C\left(t\right)}{C\left(0\right)}$$
where *C*(*t*) is the particle concentration at moment *t*, p/m^3^; and *C*(0) is the initial particle concentration, p/m^3^. In each experimental condition, the corresponding deposition rates can be regressed by the measured particle concentration series.

For the deposition rates of different ambient conditions, the analysis of influence factors was analyzed. Among the analysis methods such as correlation analysis, regression analysis, etc., the analysis of variance (also known as the F-test) can directly recognize the influence level of different independent variables on the dependent variables [24]. This method is widely used in studying airborne particle behaviors [25,26]. For this study, to quantitate the influence of the biomass’s physical properties and ambient factors on the regressed deposition rates, the F-test is very suitable. In this method, the significance (F) was used to judge whether the factors had a significant impact on the indicators, while the influence level of the factors on the indicators was compared by the effect size (η′^2^). The analysis of variance was conducted by SPSS after the regression of deposition rates.

## 3. Results and Discussion

### 3.1. Physical Property Test Results of Biomass Particles

The physical property test results of the biomass particles in different ambient conditions are shown in Table 2. It can be seen that the water absorption rates in different ambient conditions varied from 15.4% to 24.7%, while the results in the high-temperature–high-humidity condition were higher than those in the low-temperature–low-humidity condition, indicating that the weight of a single biomass particle can be greatly influenced by the ambient conditions. In fact, the water absorption of biomass particles is mainly caused by cellulose, hemicellulose, lignin, etc. [27]. It can also be seen from Table 2 that the moisture absorbed by the biomass particles accounts for no more than 4% of the total moisture amount in the chamber. Therefore, the impact of biomass water absorption on the internal air is negligible.

The observation of the biomass particles found that the original dry samples and the samples treated in the chamber had similar morphologies. The microstructure of the particles is shown in Figure 3. The shapes of the particles were mostly irregular. Some small particles were attached to larger ones. The morphology of biomass materials greatly depends on the degree of and the specific technology used for crushing. When biomass particles are smaller than plant cells (30~50 μm), the cell structures are damaged, exposing the pores in the fibrous tissue to the particle surface [28]. The exposed pores enhance the water absorption of biomass particles, while the cellulose, hemicellulose, lignin, and other substances are still stable. Therefore, the water absorption of biomass particles does not cause significant morphological changes in experimental temperatures or humidity conditions.

### 3.2. Concentration Evolution and Deposition Rate

To characterize the evolution of the biomass particle concentration, the ratio of the concentration *C*(*t*) at moment *t* to the initial concentration *C*(0) was defined as the dimensionless concentration. The final dimensionless concentrations of six particle diameter ranges after 30 min of deposition are shown in Figure 4. Among all the ambient conditions and all the diameter ranges, the final concentrations decreased significantly compared to the initial concentrations. For the same ambient condition, the reduction in the particle concentrations was positively correlated with the particle diameters. Under different conditions, the final dimensionless concentrations of the particles with diameters below 0.3 μm were between 0.36 and 0.60, while the final dimensionless concentrations of those measuring 5~10 μm were only 0.23~0.39. The reason for this is that, for a single particle, the deposition velocity increases with the particle mass under the influence of gravity and inertia. Many studies have also shown that particle size has a significant influence on deposition, which is consistent with the results obtained in this study [29,30].

The results show that under different ambient conditions, the evolutional trends in the particle concentrations are similar. Therefore, the condition with a temperature of 20 °C and a relative humidity of 65% was used as an example for the analysis of the concentration evolution, as shown in Figure 5. It can be seen that all the concentration curves for the different particle sizes decreased exponentially with time, without crossing. The two concentration curves of the smaller particles (with diameters below 0.5 μm) clearly decreased more slowly; the two curves of those measuring 0.5~2.5 μm were almost identical; and the two curves of those measuring 2.5~10 μm were slightly different.

According to the concentration curves with time, the natural deposition rates of the biomass particles with different diameter ranges under different ambient conditions regressed, as shown in Figure 6. The deposition rates ranged between 0.9 and 2.9 h^−1^, and they were positively correlated with the particle sizes. Hussein et al. [31] studied the deposition of fungal spores and abiotic particles in the range of 0.54~6.24 μm. The deposition rates obtained under the conditions of 20~28 °C and RH = 43~60% ranged between 0.01 and 0.33 h^−1^; these values were significantly lower than the results in Figure 6. By applying 0.5~20 μm polydisperse oil droplets produced by atomizing nozzles, Thatcher et al. [22] studied the effect of indoor furnishings on particle deposition at room temperature. The obtained deposition rates ranged between 0.1 and 6.8 h^−1^; these values were significantly higher than those in Figure 6. The difference between the results of this study and those in previous research shows that the deposition of biomass particles is affected by the particle characteristics and ambient environment simultaneously.

### 3.3. Variation of Deposition Rates with Temperature and Humidity

According to Figure 6, with the same relative humidity, the deposition rates of the different particle diameters tended to decrease with the increase in temperature. However, the deposition rates of the particles with diameters below 0.5 μm varied greatly with the temperature. Specifically, with the rising temperature and the constant relative humidity of 25%, the deposition rates of the particles below 0.5 μm decreased first and then increased slightly, with the minimum values occurring at 30 °C; for the relative humidity of 45% and 65%, the deposition rates below 0.5 μm increased first and then decreased with temperature, with the maximum value occurring at 25 °C. These characteristics indicate that the deposition rates of the biomass particles were mostly higher at lower temperatures. For the particles with diameters larger than 0.5 μm, the deposition rates were greatly influenced by gravity and inertia. When the temperature rises, the density of air decreases, and the dynamic viscosity of air increases. According to Stokes’ law of deposition [32], both the deposition velocity and the deposition rate of particles decrease with increases in temperature. For particles with diameters below 0.5 μm, the effect of gravity is weak due to their extremely small weight and volume, and the Brownian motion increases with the temperature, resulting in greater effects at higher temperatures.

Under the high-humidity conditions, the maximum deposition rates of the particles with diameters below 0.5 μm occurred at about 25 °C. In another study, Strategic et al. [33] examined the particle deposition in an energy-saving building and found that when the ambient temperature increased from 22 °C to 29 °C, the deposition rate of particles below 1 μm increased by three times. Xiao et al. [34] studied the particle deposition in winter houses and found that the deposition rates of the diameters ranging from 0.25~1.0 μm were significantly positively correlated to the indoor air temperature between 15 °C and 26 °C. The results of this study are different from those in the research cited above.

Regarding the influence of environmental humidity on the deposition of biomass particles, the conditions of 20 °C, 30 °C, and 40 °C were used as examples for the analysis, as shown in Figure 7. It can be seen that under the conditions of 20~30 °C, the deposition rates increased with the relative humidity, indicating that the moisture in ambient air is beneficial for the deposition of biomass particles in this temperature range. This characteristic is the same as the results from PM_2.5_ and PM_10_ reported by Kim et al. [12] and the results from aerosolized SiO_2_ particles reported by Wang et al. [35]. At 40 °C, as shown in Figure 7c, with the increase in relative humidity, the increase in the deposition rates of the large particles gradually weakened, while the deposition rates of those with diameters below 0.5 μm showed a downward trend.

Based on the physical property test and deposition rate analysis, the particle size, the ambient temperature, and the ambient relative humidity all influenced the natural deposition process of the biomass particles. For the large particles with diameters above 0.5 μm, the density grew with the water absorption amount, resulting in a positive correlation between the deposition rates and the ambient relative humidity. However, for the small particles with diameters below 0.5 μm, the Brownian motion was another non-negligible factor in the deposition, leading to higher deposition rates at lower temperatures.

### 3.4. Significance Analysis of Influencing Factors

To analyze the degree of influence of the particle size, the ambient air temperature, and the ambient air relative humidity on the deposition rates of the biomass particles, a three-factor variance analysis was conducted, with the particle sizes and ambient parameters as the independent variables and the deposition rates as the dependent variables. The results are shown in Table 3. It should be noted that the ambient temperature and relative humidity are independent variables with no interaction because they were independently controlled in the experiment. It can be seen from Table 3 that the significance of the three factors was less than 0.001, which was far less than the confidence level, α = 0.05, indicating that all three factors had a significant impact on biomass particle deposition. For the effect size, η′^2^, particle size > ambient temperature > ambient relative humidity, indicating that the particle size was the most significant influencing factor, followed by the ambient temperature and, finally, the ambient relative humidity.

In order to analyze the influence of the ambient factors on the deposition of particles with different sizes, a two-way analysis of variance was conducted. The results are shown in Table 4. The effects of the ambient temperature and relative humidity on the particle deposition varied with the particle size. The significance of the ambient temperature was less than 0.05, indicating that the impact of the ambient temperature on the particle deposition was significant for all the particle size ranges. The significance of the ambient relative humidity for the two diameter ranges below 0.5 μm was more than 0.05, while for the other particle sizes, it was less than 0.05, indicating that the ambient relative humidity had no effect on the deposition of the particles with diameters below 0.5 μm. The effect sizes of the ambient temperature were higher than those of the ambient relative humidity, which indicates that for all the diameter ranges, the influence of the ambient temperature on the particle deposition was greater than that of the ambient relative humidity. This was consistent with the results of the three-factor variance analysis. For the particles with diameters below 0.5 μm, the humidity effect was obviously lower, indicating that the influence of the ambient relative humidity on the deposition of the particles with diameters below 0.5 μm was far weaker than that of the ambient temperature.

## 4. Conclusions

The physical properties of biomass are easily affected by the ambient environment, resulting in distinctive behaviors in airborne biomass particles. In this paper, using an experimental chamber that could independently control the internal air temperature and relative humidity, the deposition characteristics of biomass particles under different ambient conditions were studied. The results show that the deposition of biomass particles is influenced by their physical properties and ambient conditions.

Biomass particles can absorb moisture from the ambient air without morphological change, resulting in a significant enhancement of their density. The water absorption rate ranges from 15.4~24.7% in ambient air with temperatures of 20~40 °C and a relative humidity of 25~65%.

The deposition rates of biomass particles are in the range of 0.9~2.9 h^−1^. Under the same ambient conditions, the deposition rates of biomass particles are positively correlated with particle size. For biomass particles with a diameter above 0.5 μm, the deposition can be increased by both the ambient temperature and the relative humidity; for biomass particles with diameters below 0.5 μm, the deposition rates are higher at lower ambient temperatures, and they have a nonlinear relationship with ambient temperature and relative humidity.

The particle size, ambient temperature, and ambient relative humidity have significant effects on the deposition of biomass particles. The order of the significance levels is particle size > ambient temperature > ambient relative humidity. For each diameter range of biomass particles, the effect of the ambient temperature on the deposition is greater than that of the ambient relative humidity. For biomass particles with diameters below 0.5 μm, the effect of the ambient relative humidity on the deposition is far weaker than that of the ambient temperature.

## Figures and Tables

**Figure 1 ijerph-20-01890-f001:**
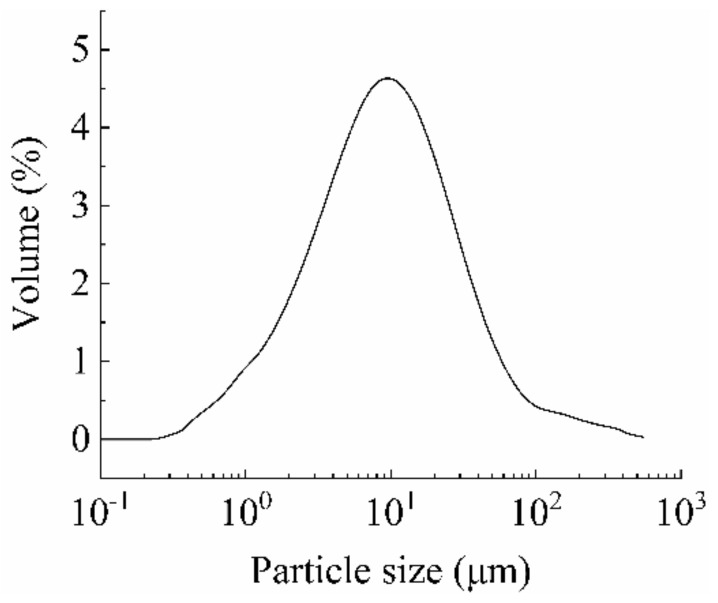
Size distribution of experimental biomass particles.

**Figure 2 ijerph-20-01890-f002:**
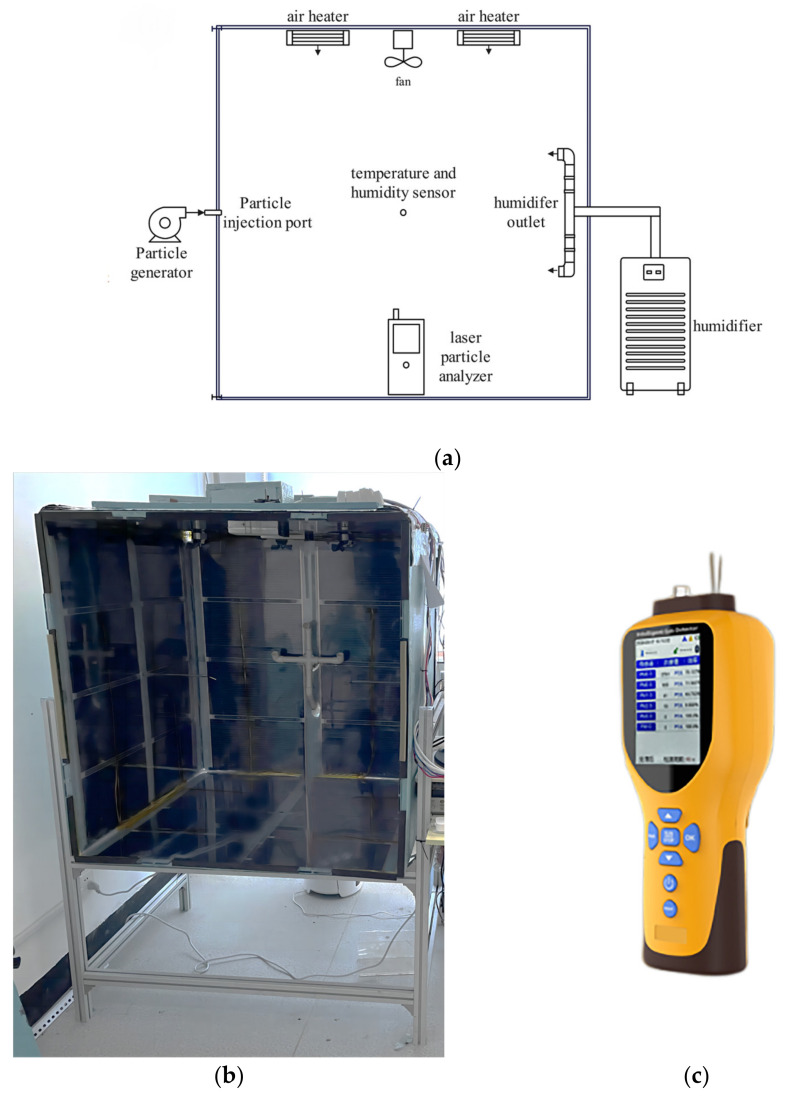
(**a**) Schematic diagram of experimental chamber. (**b**) Effect diagram of the experimental chamber. (**c**) Laser particle analyzer.

**Figure 3 ijerph-20-01890-f003:**
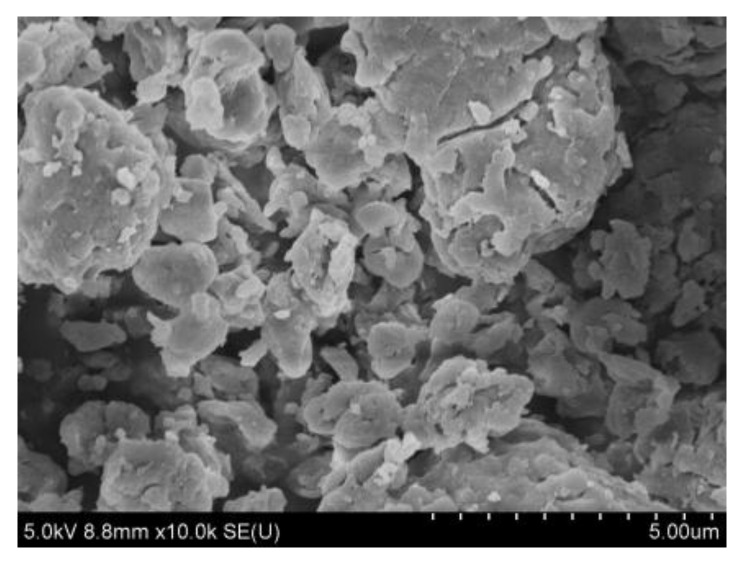
Morphologies of biomass particles (magnification of 10,000).

**Figure 4 ijerph-20-01890-f004:**
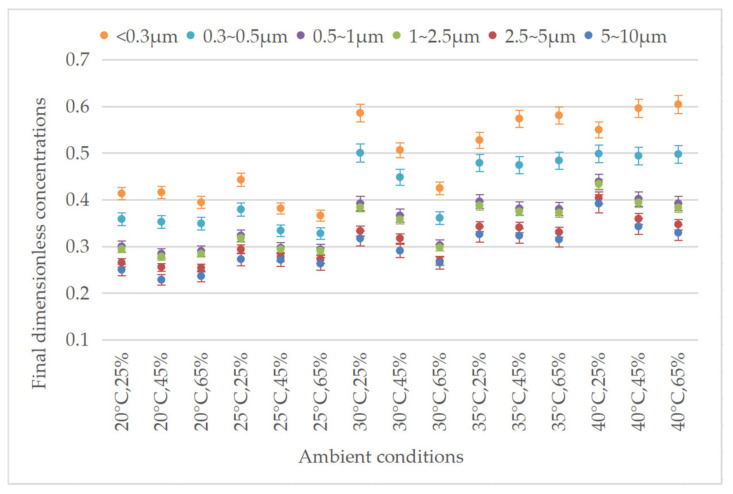
Final dimensionless concentrations of different particle sizes under different ambient conditions.

**Figure 5 ijerph-20-01890-f005:**
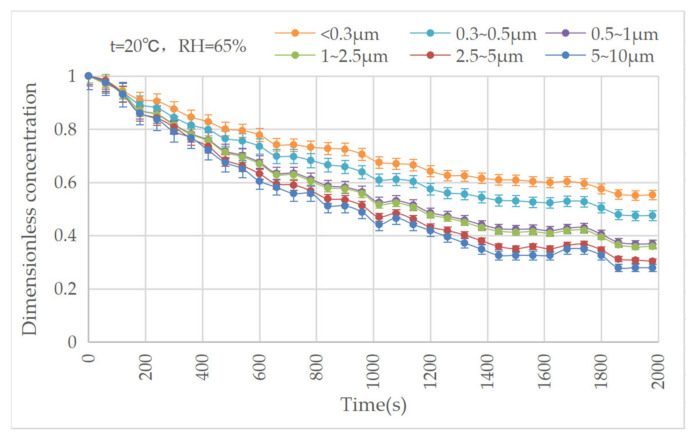
Evolution of particle concentrations in 20 °C, RH = 65%.

**Figure 6 ijerph-20-01890-f006:**
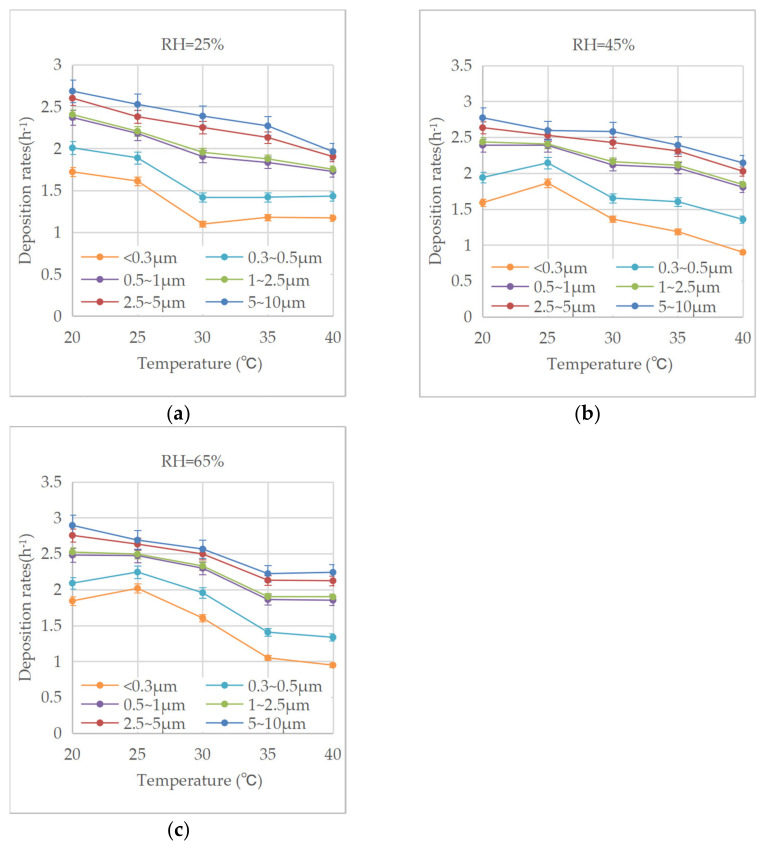
(**a**) Variation in deposition rates with ambient temperature under RH = 25%. (**b**) Variation in deposition rates with ambient temperature under RH = 45%. (**c**) Variation in deposition rates with ambient temperature under RH = 65%.

**Figure 7 ijerph-20-01890-f007:**
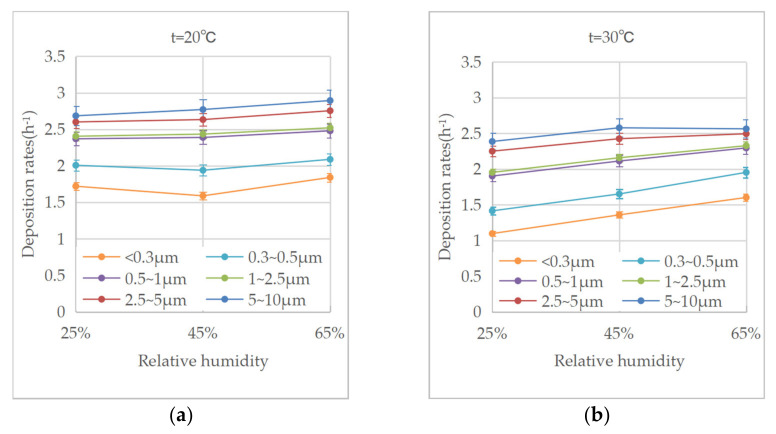
(**a**) Variation in deposition rates with ambient relative humidity at 20 °C. (**b**) Variation in deposition rates with ambient relative humidity at 30 °C. (**c**) Variation in deposition rates with ambient relative humidity at 40 °C.

**Table 1 ijerph-20-01890-t001:** Key parameters of the measuring devices.

Device	Measurement Content	Key Parameters
Colnot GT–1000 laser particle size analyzer	Particle concentration	Working temperature: 5~45 °C. Working relative humidity: ≤90%. Cumulative error: <5% with 2,000,000 particles per cubic foot (0.028 m^3^).
DWL–20E temperature and humidity recorder	Ambient temperature and relative humidity	Measurement range: −40~125 °C, 0~100% RH. Measuring accuracy: ±0.3 °C, ±3% RH.
T-type thermocouples	Wall temperature	Measurement range: −50~200 °C. Measuring accuracy: ±0.5 °C.

**Table 2 ijerph-20-01890-t002:** Water absorption rates of biomass particles.

Air Parameters in Chamber	Test Group	Dry Particle Quality (mg)	Particle Quality with Absorbed Water (mg)	Quality of Absorbed Water (mg)	Water Absorption Rate	Vapor Quality in Chamber (mg)	Proportion of Absorbed Water to Total Vapor in Chamber
20 °C, RH = 25%	1	0.926	1.069	0.143	15.4%	4.339	3.29%
	2	0.964	1.129	0.165	17.1%		3.80%
	3	0.927	1.095	0.168	18.1%		3.87%
40 °C, RH = 65%	1	0.992	1.175	0.182	18.3%	34.228	0.53%
	2	0.830	1.013	0.182	22.0%		0.53%
	3	0.810	1.010	0.200	24.7%		0.58%

**Table 3 ijerph-20-01890-t003:** Results of unrepeated three-way analysis of variance (α = 0.05).

Source	Type III Sum of Squares	df	Mean Square	F	Sig.	η′^2^
Corrected Model	18.092	11	1.645	136.591	<0.001	0.951
Intercept	373.439	1	373.439	31,013.16	<0.001	0.997
Relative humidity	0.449	2	0.224	18.635	<0.001	0.323
Particle size	11.873	5	2.375	197.211	<0.001	0.927
Temperature	5.77	4	1.442	119.794	<0.001	0.86
Error	0.939	78	0.012	——	——	——
Total	392.47	90	——	——	——	R^2^ = 0.951

**Table 4 ijerph-20-01890-t004:** Results of unrepeated two-way analysis of variance (α = 0.05).

Particle Size	Factor	Mean Square	F	Sig.	η′^2^
<0.3 μm	temperature	0.384	12.545	0.002	0.862
	relative humidity	0.026	0.856	0.460	0.176
0.3~0.5 μm	temperature	0.304	13.724	0.001	0.873
	relative humidity	0.038	1.719	0.239	0.301
0.5~1 μm	temperature	0.212	23.657	0.001	0.922
	relative humidity	0.050	5.604	0.030	0.584
1~2.5 μm	temperature	0.208	25.971	0.001	0.928
	relative humidity	0.050	6.266	0.023	0.610
2.5~5 μm	temperature	0.197	38.935	0.000	0.951
	relative humidity	0.041	8.118	0.012	0.670
5~10 μm	temperature	0.205	37.609	0.001	0.950
	relative humidity	0.034	6.320	0.023	0.612

## Data Availability

The datasets used and/or analyzed during the current study are available from the corresponding author upon reasonable request.

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
