# Peer review of "Effects of Ambient Temperature and Humidity on Natural Deposition Characteristics of Airborne Biomass Particles"

_ijerph, 2023, doi:10.3390/ijerph20031890_

Round 1

Reviewer 1 Report

Line 83: Error! Reference source not found.

Line 100: Error! Reference source not found. Please double-check if there are similar problems throughout this paper.

Line 120: Same citation problem as above.

Line 129 and 130: PM2.5 and PM10 should be in subscript

Line 131: Same citation problem as above.

Line 196: The research methodology lacks significant novelty to the study.

Line 198: The analytic method is very ordinary.

Line 204, 210, 216, 231, 248, 258, 263, 267, 274, 301, 306, 325, 328, 337: Same citation problem as above.

Line 304: Same problem as above. PM2.5 and PM10 should be in subscript.

Line 354: Should avoid using first person or “they”

Line 365: Should avoid using numbering in the conclusion section. Keep it short and concise.

Overall, there is much room for improvement in this manuscript, with huge mistakes in the citations listed above. The research methodology of this study lacks significant novelty, and the analytic method is very ordinary. Thus, I do not recommend this manuscript to be published unless major revisions is made.

Reviewer 2 Report

The paper examined the natural deposition characteristics of biomass particles in different ambient temperatures. Generally, the paper is well structured. However, I would like to point out some minor issues to improve the paper:

·        An unnecessary dot is in line 10.

·        Lines 16 and 17: Higher or lower ambient temperature and relative humidity increased the deposition ….?

·        Please add a picture from the real chamber used in this paper.

·        Please clarify why these air temperature and relative humidity values in the chamber have been chosen.

·        Line 323: Please mention three-way ANOVA. Same as in line 336 (two-way ANOVA).

·        I suggest adding a section for the discussion and the interpretation of the obtained results from the perspective of previous studies be placed in that section.

Reviewer 3 Report

In the present study, the authors used an experimental chamber that could independently control the internal air temperature and RH, and the deposition characteristics of biomass particles under different ambient conditions were studied. They found that the deposition of biomass particles is influenced by their physical properties and ambient conditions. In general, the experimental design of this study was reasonable and many experiments were conducted. The conclusions of the study were based on experiments. But some issues need to be clarified and revised by the authors.

1. Detailed parameters of the instrument, e.g. principle, accuracy, and detection line need to be described.

2. As far as I know, the principle of the GT-1000 laser particle size analyzer is based on laser, which is greatly affected by humidity. Therefore, I doubt the test results in the study.

3. The authors performed several parallel tests, but they are not shown in the figure by using the error bar.

4. We believe that before submitting the manuscript, the author should carefully check it to avoid some mistakes. For example, in lines 83, 100, and 120, Error! Reference source not found.

5. Some basic rules need the author's attention. For example, a space between numbers and units is required. In Fig. 6 and 7, the position of the sub-drawing number (a, b, c) needs to be adjusted.

In summary, considering that the authors have done a lot of work, we suggest that the manuscript needs major revision.

Round 2

Reviewer 1 Report

The authors have fixed the mistakes raised by the reviewer. I have no further comments, and the paper may be published upon the editor's decision. The English in the paper can be further improved for better reading. 

Reviewer 3 Report

Thank the authors for revising the manuscript. I think the manuscript can be accepted.